# The Effect of Fine and Coarse Recycled Aggregates on Fresh and Mechanical Properties of Self-Compacting Concrete

**DOI:** 10.3390/ma12071120

**Published:** 2019-04-04

**Authors:** Mahmoud Nili, Hossein Sasanipour, Farhad Aslani

**Affiliations:** 1Department of Civil Engineering, Bu-Ali Sina University, Hamedan 65178-38695, Iran; sasanipourhossein@gmail.com; 2School of Engineering, Hamedan University of Technology, Hamedan 65169-13733, Iran; 3Materials and Structures Innovation Group, School of Engineering, University of Western Australia, Perth, WA 6009, Australia; farhad.aslani@uwa.edu.au; 4School of Engineering, Edith Cowan University, Perth, WA 6027, Australia

**Keywords:** self-compacting concrete, recycled concrete aggregates, fresh properties, mechanical properties

## Abstract

Today, the use of recycled aggregates as a substitute for a part of the natural aggregates in concrete production is increasing. This approach is essential because the resources for natural aggregates are decreasing in the world. In the present study, the effects of recycled concrete aggregates as a partial replacement for fine (by 50%) and coarse aggregates (by 100%) were examined in the self-compacting concrete mixtures which contain air-entraining agents and silica fumes. Two series of self-compacting concrete mixes have been prepared. In the first series, fine and coarse recycled mixtures respectively with 50% and 100% replacement with air entraining agent were used. In the second series, fine recycled (with 50% replacement) and coarse recycled (with 100% replacement) were used with silica fume. The rheological properties of the self-compacting concrete (SCC) were determined using slump-flow and J-ring tests. The tests of compressive strength, tensile strength, and compressive stress-strain behavior were performed on both series. The results indicated that air-entraining agent and silica fume have an important role in stabilization of fresh properties of the mixtures. The results of tests indicated a decrease in compressive strength, modulus of elasticity, and energy absorption of concrete mixtures containing air entrained agent. Also, the results showed that complete replacement (100%) with coarse recycled material had no significant effect on mechanical strength, while replacement with 50% fine recycled material has reduced compressive strength, tensile strength, and energy absorption.

## 1. Introduction

Nowadays, it is well known that preparing natural aggregates for concrete production—due to carbon dioxide consumption—is a severe threat of environment and should be logically minimized. On the other hand, the construction and demolition waste (CDW) in countries is growing. In this regard, little more than half a billion tons of annual waste generated in Europe, this amounts to 325 million and 77 million tonnes for the US and Japan respectively. Considering that China and India are now producing and using over 50% of the world’s concrete, as development continues their CDW will be important too [1,2]. It was reported by UEPG (European Aggregates Association) that aggregates consumption for producing of concrete was 2.5 billion tonnes in 2013. Figure 1 illustrates the tonnes of aggregates produced per capita by countries. As shown, Finland produced about 16 tonnes per capita which ranks as the top producer amongst EU countries [3]. 

Despite restrictions for preparing of aggregates, demands for new concretes like self-compacted concrete (SCC), due to advantages of manufacture and high efficiency at the final stage, have been expanded around the world [4,5,6]. The main benefits of SCC come back to the fact that it can be easily poured into the formwork of highly reinforced structures without any vibration. Decreasing construction time, maximizing the freedom of design work, and improvement in quality of product and working environment are some other advantages of this feature. Nowadays, contractors are encouraged to use industrial by-products and also CDW, like recycled concrete aggregate (RCA), as a partial replacement for aggregates in concrete mixtures [7,8]. It is clear, due to mortar adhering to recycled aggregates, that the fresh and hardened properties of concrete made with these aggregates were negatively affected. For instance, the workability, density, as well as compressive strength, tensile strength, and modulus of elasticity (MOE), were reduced [9]. The density of CDW, due to the presence of lower density remaining cement mortar particles attached to the aggregate, was lower than that of natural aggregate [10]. However, when the adhered mortar contact in RCA is less than 44%, the recycled aggregate can be used in structural concrete elements [9]. Adding CDW aggregate led to a decrease of compressive strength by 12 and 25 %, when 25–30% [11,12] or 100 % NA was substituted by CDW aggregate [13,14,15]. However, no significant effect was seen when the coarse or fine recycled aggregate was applied to substitute up to 30 % of coarse NA [16,17,18,19,20] or 20 % of fine NA [21]. 

It was also reported that the slump of recycled aggregate concrete (RAC), due to higher water absorption, angularity, and rough texture of CDW, was lower than that of conventional ones [22,23]. For this reason of prevention of reducing of workability in RAC, use of recycled aggregates in saturation surface drying or spraying on recycled aggregates by sprinkler system is recommended [12,20,24,25]. Corinaldesi and Moriconi (2011), studied the effect of both fine and coarse RCA and also some mineral additives on SCC properties. They showed that rubble powder made better concrete flowability and flow-segregation resistance in comparison with fly ash and limestone powder [26].

Contradictory results were reported for hardened properties of recycled aggregate concrete. Some researchers showed that the compressive strength of concrete was increased as RCA was used as replacement for normal aggregates. These improvements were attributed to the rough texture in recycled aggregates which provided better bonding and interlocking between the cement paste and the recycled aggregates [12]. 

However, recycled concrete aggregates may adversely impact the mechanical properties of recycled aggregate concrete. It is evident that the reduction of compressive strength of RACs compensates by adjusting the water to cement ratio, changing the procedure of mixing, treating the aggregate, and using a mineral addition [22]. On the other hand, the result of different research declared that the substitution ratio of natural aggregate by CDW aggregate had no negative impact on the splitting tensile strength compared to that for compressive strength [27], which may be due to the high bond strength between RCA and cement paste [28]. 

Modulus of elasticity of recycled aggregate concrete (RAC) specimens usually is less than that of conventional concrete, and it declines as the content of CDW aggregate in concrete grows. 

The causes for the decline in concrete’s MOE were due to the loss of concrete stiffness and porosity and also aggregate-cement paste bonding [22]. Most authors declared that RCA had a more negative impact on MOE than those for compressive strength [29,30]. The trend of curves of stress-strain are similar for both recycled and conventional concretes, however, a slight shift—due to differences MOE and ultimate strain—was observed in recycled ones. The shift is significant when the substitution rate is high. The longitudinal strain of the recycled concretes goes up with the percentage of recycled coarse aggregate used [30]. 

However, there are two main methods to enhance the properties of recycled concrete aggregates in order to improve concrete properties; 1) removing the adhered mortar, and 2) strengthening the adhered mortar. Several methods are used for removing the adhered mortar on RCAs, including physical and chemical treatment. Mechanical grinding, pre-soaking in water, and pre-soaking in acid are the common methods for removing the adhered mortar [31]. Improving the adhered mortar is another way to enhance the quality of RCAs. In this method by coating mineral admixtures on RCAs, such as silica fume, the interface transition zone can improve [32]. The high level of fineness and practically spherical shape of silica fume results in good cohesion and improved resistance to segregation. It is also very effective in reducing or eliminating bleed and this can give rise to problems of rapid surface crusting. This can result in cold joints or surface defects if there are any breaks in concrete delivery and also present difficulty in finishing the top surface (Cited by EFNARC) [33]. Additionally, it is well understood that silica fume, due to high pozzolanic activity, is an inevitable component when producing high strength concrete; silica fume effectively improves the structure of the transition zone, eliminates the weakness of the interfacial zone, reduces the number and size of cracks (cited by Nili et al.) [34].

Grdic et al. (2010) used high-quality rubble RCA by 0%, 50%, and 100% replacement for natural coarse aggregate in SCC mixtures. The results showed a minor loss in strength. When RFA was applied to substitute sand, the compressive strength was negatively influenced [35]. Fakitsas et al. (2012), surveyed the influence of internal curing applying saturated RA in SCC. All aggregates were plunged in water for three days and then surface dried for 12 hours before being used in the concrete mixture. The results declared that RA in SCC had shown to have a higher compressive strength compared to NA in SCC at ages of 28 and 90 days. This increase is attributed to internal curing of concrete by saturated RCA and therefore, utilization of saturated RCA may compensate for some of its other defects with strength development [36].

## 2. Research Significance

The experimental study as outlined in this paper aims to promote the use of sustainable forms of structural self-compacting concrete incorporating recycled concrete aggregates and develop information on its fresh and hardened mechanical properties of SCC. In this study, the effect of recycled aggregates in addition to air-entrained admixture and silica fume on the properties of self-compacting concrete have been investigated. The aim is to reduce waste, reduce natural aggregate consumption in the concrete industry, and promote the widespread use of self-compacting concrete.

## 3. Experimental Design and Materials

### 3.1. Materials and Mix Proportions

Ordinary Portland cement (ASTM Type 1), and silica fume were used in this work. The physical properties of both natural and recycled aggregates are given in Table 1. Limestone powder was also used to modify the viscosity of the SCC mixtures. The specific gravity of the limestone powder was 2.7. The amount of adhered mortar in RCA was measured using the methodology given by [9] and the percentage of adhered mortar was calculated using Equation (1). This method consists of soaking in water and heating of the aggregate.
(1)$$M_{c}\;(\%)=\frac{m_{i}-m_{f}}{mi}\times 100\;$$
where *M_c_* (%) is the amount of adhered mortar in RCA, $m_{i}$ is the total mass of the RCA sample; *m_f_* is the mass of the same NCA (natural coarse aggregates) sample.

As given in Table 1, the water absorption of coarse and fine recycled aggregates is 316% and 109% more than that for natural ones, respectively. Furthermore, the abrasion resistance of recycled aggregate is 164% lower than the natural coarse aggregate. These results, which are due to the adhered mortar, are in agreement for the results obtained by other researchers [37,38], and are most likely to conform to EN1097-6, reporting the physical properties of natural and recycled aggregates [39]. The gradations of coarse and fine aggregate and limestone powder are shown in Figure 2.

Two types of superplasticizer agent with the commercial name of WBK50 and WRM (LG) were combined and used to adjust the workability of the self-compacting concretes. A range of values for an adequate slump flow is 550–850 mm [33].

### 3.2. Mix Properties and Procedure

In this research, two series of SCC mixtures with water-(cement + Sf) ratios of 0.44 and cementitious material contents of 418 kg/m^3^ were prepared and labeled as I and II, respectively. Table 2 shows the mix proportions of the mixtures. As given, the fine and coarse recycled aggregate was used as 50 and 100% by volume replacements of natural aggregate, respectively. In Series I, the air entraining agent (AEA), (0.03% by weight of cement) was used in concrete mixtures prepared with recycled aggregates. In Series II, silica fume was also added as a cement replacement (8% by weight). Totally, eight mixtures were considered containing reference samples (S and SS). The specimens with 100% (by volume) coarse RCA replacement were labeled as S100C and SS100C in Series I and II, respectively. The other concretes in both series, which were labeled as S50f and SS50f, contained 50% (by volume) fine RCA as a partial replacement. The concretes which were produced with both fine and coarse RCAs were known S100C50f and SS100C50f. The mixing procedure was designed by trial and error as follows: the cement, limestone powder, fine aggregate, and half of the mixing water were mixed initially for 1 min, and then the remaining mixing water was mixed up with superplasticizer, and a little of it was added to the mixture, stirred for 3 min. Finally, the coarse aggregate and the rest of the water- superplasticizer were added and mixed for 3 min. It is noted that, in concrete mixture Series I, AEA was added at the last step. In Series II, silica fume and $\frac{1}{3}$ mixing water was mixed and then added to concrete mixtures. The fine and coarse recycled aggregates were pre-soaked for 24 h before being used in the mixtures. Pre-soaking is considered an effective way to separate impurities and obtain higher quality RCA [31].

### 3.3. Test Program and Procedures

#### 3.3.1. Slump Flow and J-Ring

The tests of slump flow and J-ring were carried out on the self-compacting concrete specimens according to EFNARK standard.

#### 3.3.2. Compressive and Splitting Tensile Strength Testes

The splitting tensile strength test was performed at 28 days on 100 × 200 mm^2^ cylindrical specimens. Cubic specimens of 100 mm were used for determination of compressive strength test which was performed at ages of 7, 28, and 91 days. 

#### 3.3.3. The Stress-strain Relationship

Compressive strength stress-strain tests were performed at the age of 28 days on 100 × 200-mm cylindrical specimens. For this purpose, three capped samples were used in order to use their mean to plot stress-strain curves. Figure 3 shows the setup instruments for stress-strain measuring: compression machine of ADR 2000 KN (ELE Company, England, UK), strain gauge, and data logger. 

The stress-strain of the capped cylinder samples, at the age of 28 days, was measured via the system arranged in Figure 3. The stress-strain curve of every mixture was the average of three tested samples.

## 4. Test Results and Discussion

This section presented the experimental results obtained in the laboratory and discussion for properties of eight concrete mixtures.

### 4.1. Fresh Concrete Results 

Several trial and error efforts were made for preparing the convenient SCC mixtures. It was found that to avoid bleeding and segregation in concrete mixtures prepared with recycled aggregates, AEA and silica fume must be used (Figure 4a–c). This important finding is surely an effective guidance contractors to prepare SCC mixtures made by fine and coarse RCAs. 

The results of the slump flow and J-ring tests are given in Table 3. The results indicate that use of the recycled aggregates in series I have reduced the slump flow. It is shown that replacing 100% of coarse and 50% of fine recycled aggregates with normal ones, S100C and S50f, led to 20 mm reduction of slump flow compared to the reference sample S. Simultaneous use of recycled fine and coarse aggregate in the S100C50f has reduced the slump flow by 50 mm. However, the higher amount of superplasticizer in Series II compensated the reduction of slump flow of the coarse recycled concretes. The replacement of 50% fine aggregate in the SS50f suggests that fine RCA also has reduced slump flow. Increasing the superplasticizer content in the SS100S50f led to an increase of the slump flow without segregation. As given, the flow of all mixtures in both series satisfied the limits of EFNARC standard [33]. Figure 5a,b show the spread view of the mixtures after the slump flow test. As shown, no segregation and bleeding has occurred in the concrete mixtures.

The results of the J-ring test showed that, due to the presence of rebar in this test, the measured diameter had been decreased compared to the slump flow test in all mixture (Figure 6a,b) [1]. According to the results of Table 3, the height differences in the J-ring test for all mixtures are less than 10 mm. It is also shown that there was no segregation and bleeding in any of the mixtures that were tested by the J-ring test (Figure 7).

### 4.2. Properties of Hardened Concrete

#### 4.2.1. Compressive Strength

The results of compressive and tensile strength are shown in Table 4. As given the compressive strength of S100C, due to the replacement of 100% coarse RCA, diminished by 38.5%, 30.7%, and 25.2% compared to reference sample S, respectively, at the age of 7, 28, and 91. Moreover, the replacement of 50% of fine RCA in S50f has reduced compressive strength by 27.1%, 24%, and 30.6% at the age of 7, 28, and 91 days, respectively. The results indicate that the simultaneous use of fine and coarse RCA in S100C50f led to a reduction of compressive strength by 22.7% at the age of 91 days (Figure 8a). It is noteworthy that air entraining agent is normally responsible for compressive strength reduction; however, the rule of this admixture on strength of recycled aggregate concrete is not so clear. Similar results were obtained by other researchers. They found that the main reasons for strength reduction were attributed to the higher porosity of recycled aggregate and the weak aggregate-matrix interface bond between recycled aggregate and mortar [40]. On the contrary, in series II, the compressive strength of SS100C increased by 6.5%, 14%, and 28%, compared to SS sample at the ages of 7, 28, and 91 days, respectively. This result revealed the advantage effect of silica fume when used as a replacement for cement. The other researchers also showed that utilization of mineral admixtures—like fly ash, silica fume, and ground granulated blast furnace slag—in the mix proportions led to an increase of compressive strength [41,42,43]. The results also showed that the compressive strength of SS50f mixture at the ages of 7, 28, and 91 days was decreased by 31.6%, 29.9%, and 22.1%, respectively. The result declared that the compressive strength of SS100C50f, which contained simultaneously coarse and fine recycled aggregates, reduced about 26.6% at the age of 91 days (Figure 8b).

#### 4.2.2. Splitting Tensile Strength

The results of the tensile strength test at the age of 28 days are illustrated in Figure 9. As shown the use of recycled aggregates had no significant effect on tensile strength in Series I. Replacing 50% of the fine RCA in the S50f has led to a slight reduction of 7%. In Series II with a 100% replacement of coarse RCA in SS100C, the tensile strength was reduced by 5.2%. In some studies, it is reported that due to improvement in ITZ (interface transition zone) the recycled aggregates can affect a negligible reduction in tensile strength [44,45]. However, the tensile strength of SS50f and SS100C50f specimens decreased by 31% and 27.2%, respectively. As shown, the air entrained agent had no negative impact on the tensile strength of the mixtures. This is an important issue which more study is necessary for fully understanding the reasons. 

#### 4.2.3. Compressive Stress-strain Behavior

The stress-strain test results at 28 days are given in Table 5. Some properties, including the proportional strain test (peer), the maximum stress and the final strain, the energy absorbed in the intervals $\varepsilon_{c}:\left(0-\varepsilon_{c}^{\prime }\right)$ and $\varepsilon_{c}:\left(0-\varepsilon_{u}^{\prime }\right)$ and the MOE are also given in Table 5. The results showed that in all recycled concrete mixtures, with the replacement of recycled aggregates, the energy absorption in the interval $\varepsilon_{c}:\left(0-\varepsilon_{c}^{\prime }\right)$ has decreased compared to the reference samples (Figure 10a). In the interval $\varepsilon_{c}:\left(0-\varepsilon_{u}^{\prime }\right)$, the energy absorption decreased by replacing recycled aggregates except for S100C (Figure 10b).

In Figure 11a, the stress-strain relationships of Series I specimens are shown. By replacing recycled aggregates, the modulus of elasticity (MOE) has decreased compared to the reference sample S. By replacing 100% coarse RCA, MOE has decreased by 14% and by replacing 50% of the fine RCA, the MOE has declined by 22%. The MOE was also reduced by 10% in S100C50f, by combining fine and coarse recycled aggregates.

In Figure 11b, the stress-strain curves of Series II is depicted. The use of coarse RCA in SS100C reduced the MOE by 10%, while utilization of fine RCA in SS50f led to a 23% increase of MOE. Also, the composition of recycled aggregates caused a sharp decline in the MOE by 34% compared to the SS. The failure pattern of mixtures including recycled aggregate in Series I is shown in Figure 12a. Considering the failure pattern in this series, the pattern of cracks in the mixtures containing recycled aggregates is vertical while the crack pattern in the reference sample is diagonal with an angle of approximately 45 degrees. These differences in patterns of failure clarify that cracks passed within the recycled aggregates. In Series II, which silica fume was used in the mixtures, similar crack patterns were observed in both reference and recycled specimens. This pattern revealed that silica fume compensated the weak effect of the recycled aggregates as a replacement for natural aggregates (Figure 12b).

## 5. Conclusions

According to the research carried out in this study, the following results are presented as research achievements:(1)Utilization of recycled aggregates reduced the slump flow, but the use of higher amounts of superplasticizer could prevent reduction of slump flow. Also, the use of recycled aggregates in self-compacting recycled concrete showed acceptable passing ability in the J-ring test.(2)Replacement of recycled aggregates in the mixtures including air-entraining admixture reduced compressive strength. However, no significant reduction of splitting tensile strength was observed.(3)Replacement of 100% coarse recycled concrete aggregates did not affect the compressive and tensile strength of the silica fume mixtures. Where the fine recycled aggregate negatively affected the compressive strength and tensile strength of the RAC ones.(4)The recycled aggregates in concretes containing air-entrained admixture cause a decrease in compressive strength and modulus of elasticity. Substitution of 100% coarse RCA in silica fume specimens had no significant effect on the shape of the stress-strain curve, but the fine recycled aggregates caused a decrease in the energy absorption and compressive strength of the specimens.(5)Crack inspection of the recycled specimens after compressive strength testing showed vertical cracks which revealed the failure of the recycled aggregate. Introducing silica fume into the recycled specimens changed the inclined crack pattern with an angle of approximately 45° and similar to those for the crack patterns of the reference samples.(6)The present obtained results regarding utilization of RCA, as a partial replacement for natural aggregates, may lead to the fact that recycled aggregate concrete is an environmentally friendly material.

## Figures and Tables

**Figure 1 materials-12-01120-f001:**
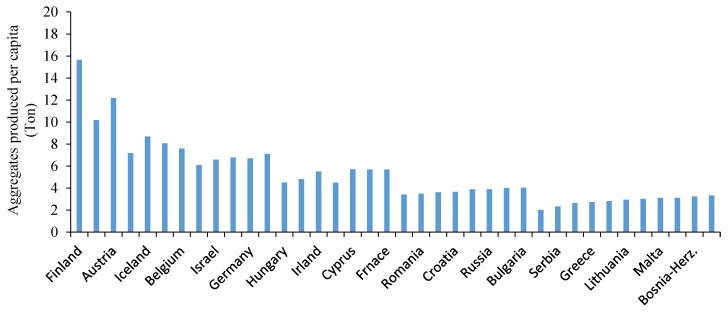
Aggregates production (tonnes) per capita by EU countries [3].

**Figure 2 materials-12-01120-f002:**
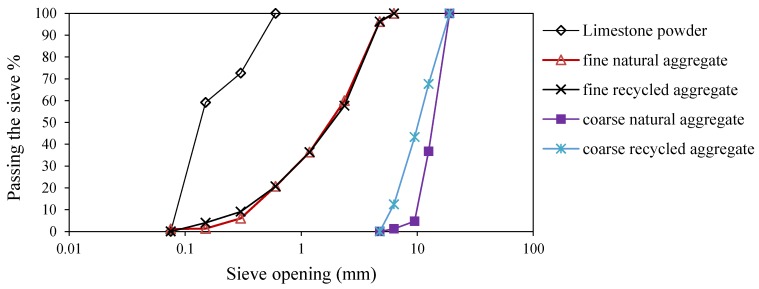
Gradations of coarse and fine aggregate and limestone powder.

**Figure 3 materials-12-01120-f003:**
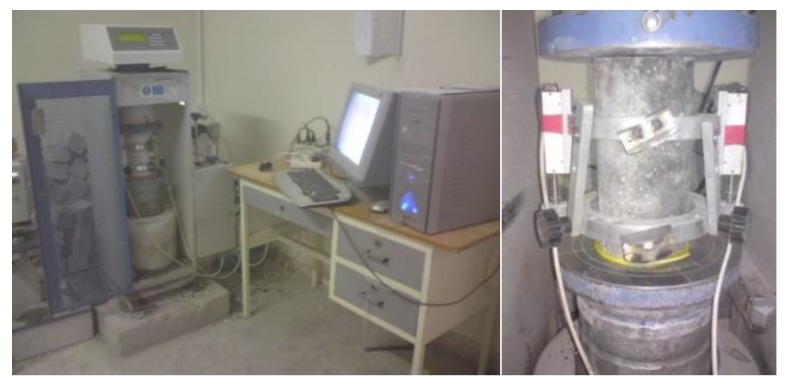
Compressive strength stress-strain tests apparatus.

**Figure 4 materials-12-01120-f004:**
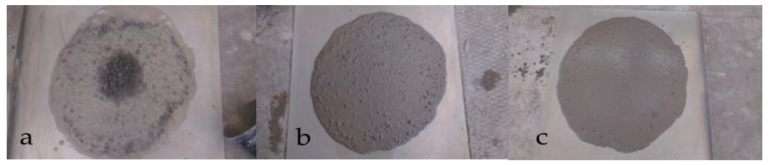
Concrete mixtures prepared with recycled aggregates: (**a**) concrete mixture without AEA and Sf; (**b**) concrete mixture with AEA; and (**c**) concrete mixture with Sf.

**Figure 5 materials-12-01120-f005:**
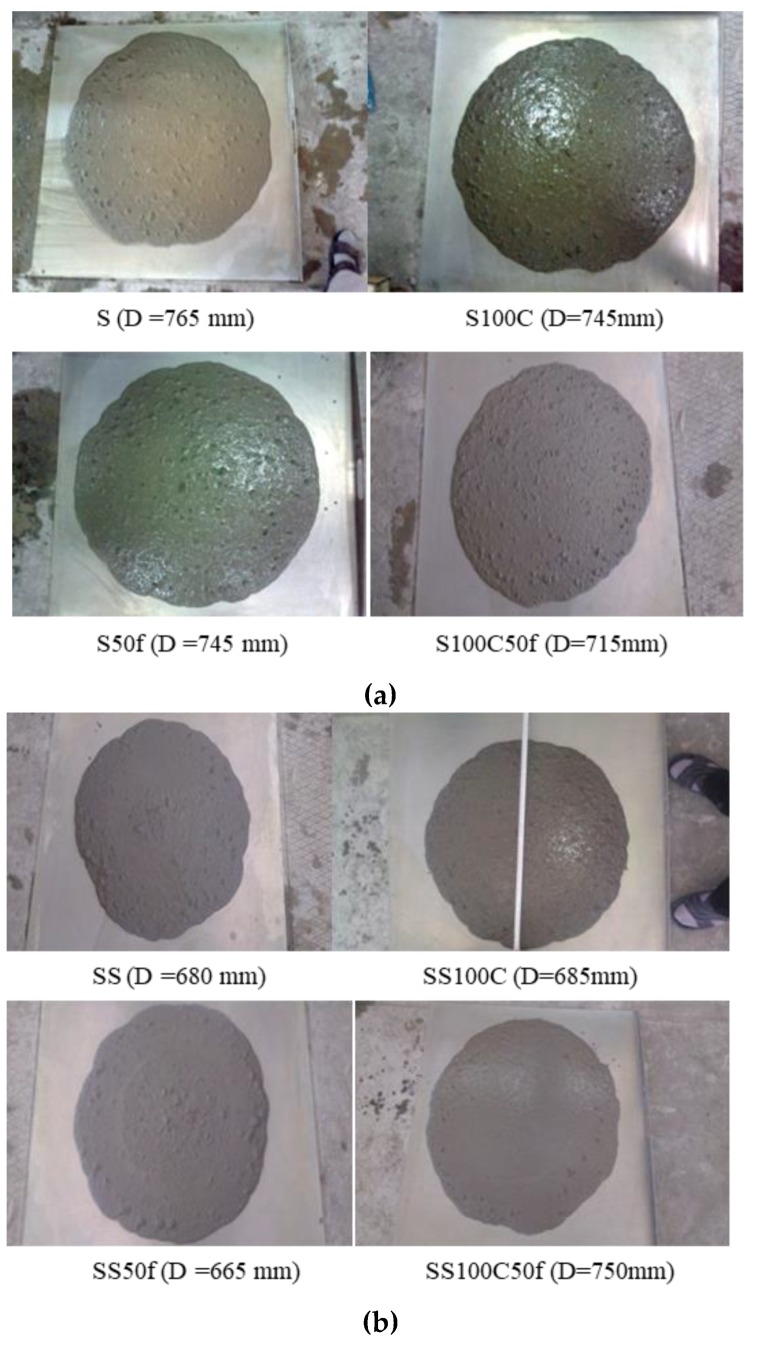
Slump flow tests (**a**) concrete mixtures series I, and (**b**) concrete mixtures series II.

**Figure 6 materials-12-01120-f006:**
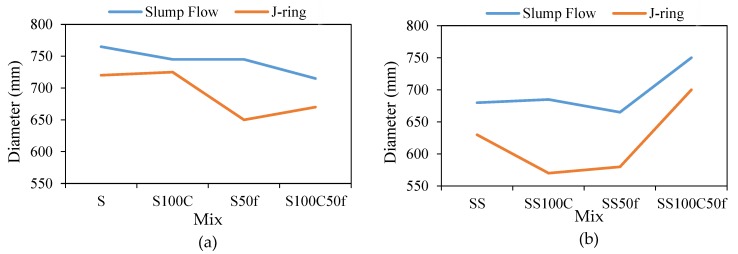
Slump test flow diameter and J-ring flow diameters (**a**) series I, and (**b**) concrete mixtures series II.

**Figure 7 materials-12-01120-f007:**
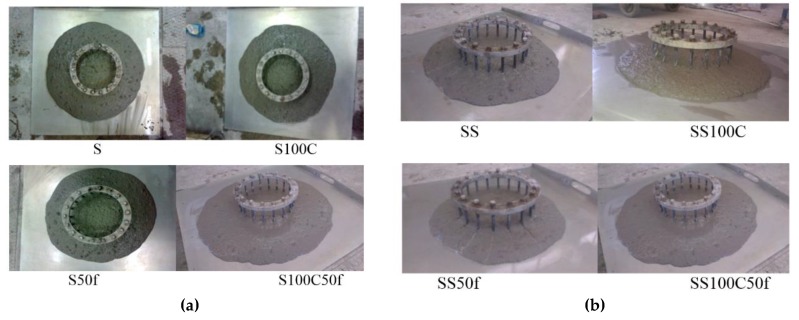
J-ring test (**a**) concrete mixtures series I, and (**b**) concrete mixtures series II respectively.

**Figure 8 materials-12-01120-f008:**
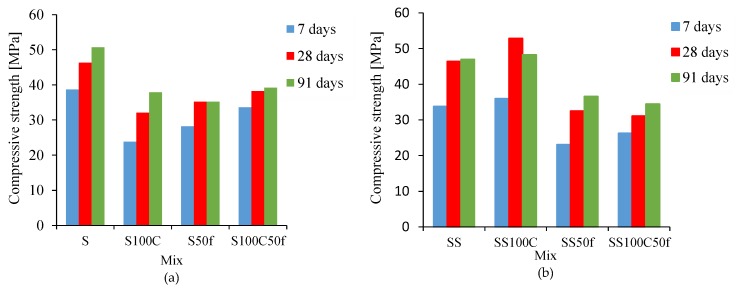
Compressive strength results at 7, 28, and 91 days of (**a**) Series I and (**b**) Series II.

**Figure 9 materials-12-01120-f009:**
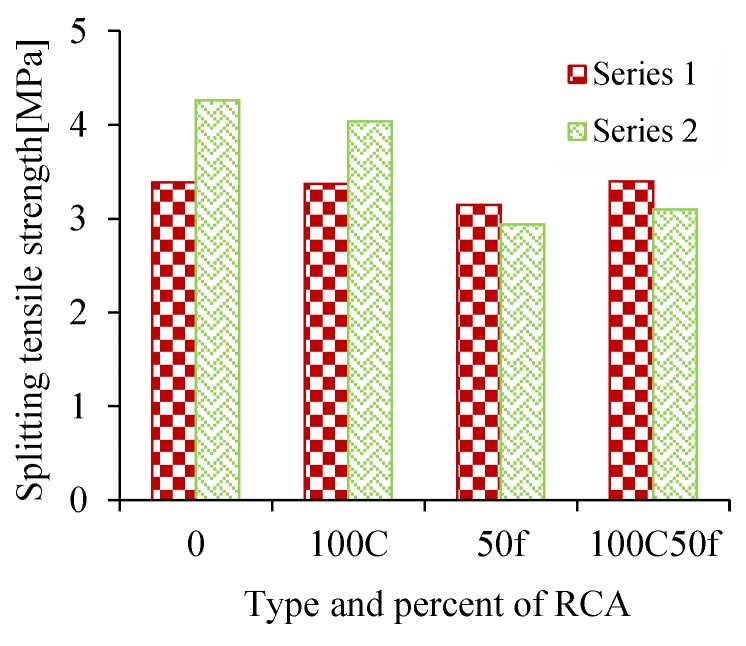
Tensile strength results at 28 days.

**Figure 10 materials-12-01120-f010:**
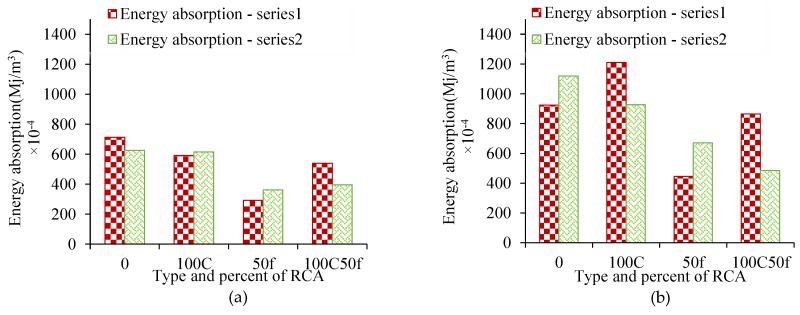
Energy absorption in the intervals, (**a**) $\varepsilon_{c}:\left(0-\varepsilon_{c}^{\prime }\right)$, and (**b**) $\varepsilon_{c}:\left(0-\varepsilon_{u}^{\prime }\right)$.

**Figure 11 materials-12-01120-f011:**
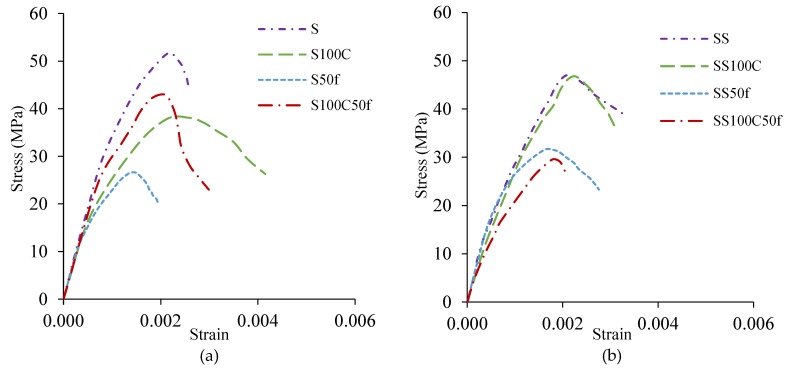
Compressive stress-strain curves for (**a**) Series I, (**b**) Series II.

**Figure 12 materials-12-01120-f012:**
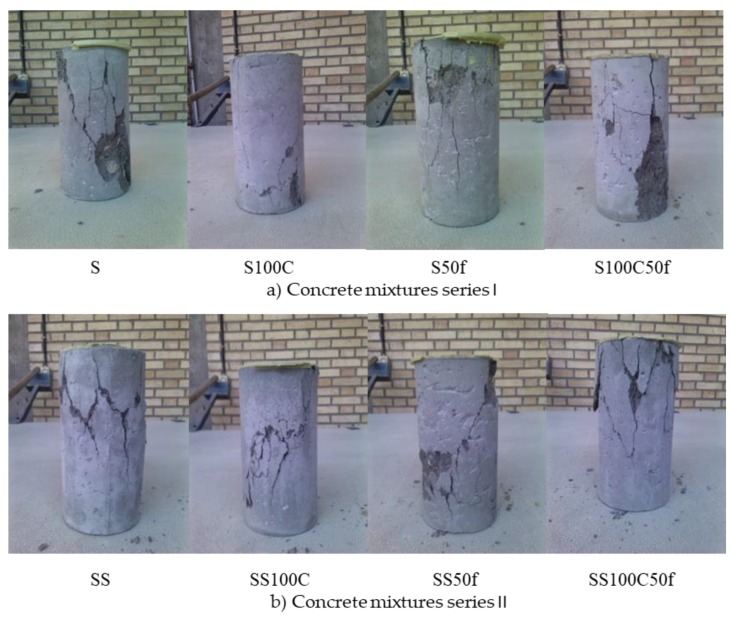
Failure pattern of mixtures in compressive stress, (**a**) Series I and (**b**) Series II.

**Table 1 materials-12-01120-t001:** Physical properties of NA and RCA.

Type of Aggregate		Specific Gravity (g/cm^3^)	Maximum Size (mm)	Fineness Modulus	Water Absorption (%)	Los Angeles Abrasion (%)	Adhered Mortar (%)
Coarse	Natural	2.66	19	-	1.21	13.9	-
	RCA	2.52	19	-	5.04	35.9	32.15
Fine	Natural	2.53	-	3.79	3.70	-	-
	RCA	2.46	-	3.76	7.76	-	-

**Table 2 materials-12-01120-t002:** Mix proportions of the concrete.

Mix Code	Series	W/(C + Sf)	W	Cement	Sf	FNA.	CNA.	FRA.	CRA	LP	SP (%)	AEA (%)
			(kg/m^3^)									
S	I	0.44	184	418	-	1172	335	-	-	167	0.8	-
S100C					-	1160	-	-	331	166	0.9	0.03
S50f					-	580	332	580	-	166	0.9	0.03
S100C50f					-	574	-	574	328	164	0.9	0.03
SS	II	0.44	184	385	33	1172	335	-	-	167	0.9	-
SS100C					33	1160	-	-	331	166	1.0	-
SS50f					33	580	332	580	-	166	1.1	-
SS100C50f					33	574	-	574	328	164	1.2	-

Sf: silica fume, FNA: fine natural aggregates, CNA: coarse natural aggregates. FRA: fine recycled aggregates, CRA: coarse recycled aggregates, LP: limestone powder. SP: superplasticizer, AEA: air-entraining agent.

**Table 3 materials-12-01120-t003:** Fresh properties test results.

Mix Code	Series	Sp (%)	Slump Flow (mm)	T50 (s)	T Final (s)	J-ring (mm)	J-ring Height Difference (mm)	Whether Conforms to EFNARC [33] Guidelines?
S	I	0.8	765	3	31	720	9	Yes
S100C		0.9	745	3	32	725	2	Yes
S50f		0.9	745	2	33	650	5	Yes
S100C50f		0.9	715	4	55	670	8	Yes
SS	II	0.9	680	2	19	630	5	Yes
SS100C		1	685	2	23	570	3	Yes
SS50f		1.1	665	3	25	580	5	Yes
SS100C50f		1.2	750	2	37	700	2	Yes

**Table 4 materials-12-01120-t004:** Mechanical properties of self-compacting concrete.

Mix Code	Series	Compressive Strength (MPa)			Splitting Tensile Strength (MPa)
		7 Days	28 Days	91 Days	28 Days
S	I	38.7	46.3	50.7	3.4
S100C		23.8	32.1	37.9	3.4
S50f		28.2	35.2	35.2	3.2
S100C50f		33.6	38.2	39.2	3.4
SS	II	33.8	46.4	47	4.3
SS100C		36	52.9	48.3	4.1
SS50f		23.1	32.5	36.6	2.9
SS100C50f		26.3	31.8	34.5	3.1

**Table 5 materials-12-01120-t005:** Compressive stress-strain curve test results.

Mix Code	Series	$\varepsilon_{c}^{\prime }\;\;\times \;10^{-3}$	Energy Absorption × 10^−4^(MJ/m^3^) $\varepsilon_{c}:\left(0-\varepsilon_{c}^{\prime }\right)$	$\varepsilon_{u}^{\prime }\times \;10^{-3}$	Energy Absorption × 10^−4^ (MJ/m^3^) $\varepsilon_{c}:\left(0-\varepsilon_{u}^{\prime }\right)$	Modulus of Elasticity (GPa)
S	I	2.18	713	2.61	924	38.8
S100C		2.33	591	4.17	1210	33.3
S50f		1.69	292	2.33	446	30.3
S100C50f		2	540	3	865	34.9
SS	II	2.18	626	3.34	1120	32.3
SS100C		2.23	615	3.09	928	29.1
SS50f		1.68	362	2.77	671	39.8
SS100C50f		2.17	396	2.48	485	21.4

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
