# Peer review of "The Effect of Fine and Coarse Recycled Aggregates on Fresh and Mechanical Properties of Self-Compacting Concrete"

_materials, 2019, doi:10.3390/ma12071120_

Round 1

Reviewer 1 Report

This paper presents an experimental investigation of the effect of fine and coarse recycled aggregates on the fresh and mechanical properties of self-compacting concrete (SCC). In the study, two series of SCC mixes have been prepared. In the first series, fine and coarse recycled concrete aggregates (RCA), respectively with 50% and 100% replacement; an air entraining agent was used in the mixture. In the second series, fine RCA (with 50% replacement) and coarse RCA (with 100% replacement) were used with silica fume. The following comments are suggested:

1.          The authors should cite more recent publications in their literature review.

2.          The citations of references in the text should follow the provisions of the journal.

3.          In Figure 2, the horizontal coordinates should indicate the unit, and should also indicate which type of aggregate the curves represent.

4.          In lines 135-136: “…Totally ten mixtures were considered containing reference samples (S and SS)....” But Table 2 has only 8 mixtures.

5.          In lines 146-147: “…The recycled aggregates were pre-wetted before using to eliminate the high porosity effect on the mixture.” What is the pre-wetting degree of RCA? Please explain further.

6.          In lines 171-172: “This section presented the experimental results obtained in the laboratory and discussion for properties of ten concrete mixtures.” But Table 2 has only 8 concrete mixtures.

7.          In lines 189-190: “…According to the results of Table 3, the height differences in the J-ring test for all mixtures are less than 10 mm.....” The provisions of the J-ring test eligibility criteria should be stated.

8.          According to the mix proportions of Series I in Table 2, the experimental group was mixed with an air-entraining agent, which may affect its compressive strength. It is not easy to understand whether the decrease in strength is caused by the recycled concrete aggregates or the air-entraining agent. Authors should give more explanations for this.

9.          In Table 4, the significant digit of compressive strengths should be consistent.

10.      In lines 230-231: “…As shown the use of recycled aggregates had no significant effect on tensile strength......” In lines 234-235: “…The results indicate that the replacement of coarse RCA had no considerable effect on the tensile strength of both series.” These sentences apply only to Series I and should be rewritten.

11.      The replacement of coarse RCA had no considerable effect on the tensile strength of Series I. But the compressive strength and modulus of elasticity were not the same, why? Authors should give more explanations for this.

12.      The concrete strain ec in the text, c should be represented by subscript.

13.      What is the age of the concrete specimens in Figure 10? Their compressive strengths should be consistent with Table 4. Authors should give more explanations for this.

14.      All codes and standards used should be listed in the references.

15.      Reference should follow the style of this journal.

Author Response

Response to Comments from Reviewer 1

Comment 1:

Some new published papers were added to part Introduction.

Comment 2:

The citation of the references was prepared based on the Journal format.

Comment 3:

The type of aggregates and the captions were shown.

Comment 4:

The number of mixtures was edited.

Comment 5:

In lines 166-168: the method of pre-soaking and advantage of this procedure was added.

Comment 6: 

The number of mixtures was 8 and it was edited in the body of the paper.

Comment 7:

One column was inserted at the right of Table 3 to show EFNARC requirements.

Comment 8:

In line 267 an explanation was added to assess how entrained agent may affect the compressive strength of RAC mixtures.

Comment 9:

In Table 4, the significant digit of compressive strengths values was edited.

Comment 10:

In lines 292 -300 the paragragh was edited and two references were added to clarify the obtained results.

Comment 11:

In line 292 it was emphasized that the tensile strength of RAC was no adversely affected by air entrained agent.

Comment 12:

The concrete strain was edited based on the comment.

Comment 13:

The age of the specimens was 28 days.  The reason is due to the specimens types which are different in compressive strength test and also stress- strain test.

Comment 14:

All codes and standards were listed in references. 

and,

Comment 15:

Reference followed the style of the journal.

Thank you very much for your consideration.

Best Regards,

Mahmoud Nili

Reviewer 2 Report

This paper presents results of experimental studies on feasibility of utilizing recycled aggregates as a replacement for new aggregates in concrete. This study presents good insights on the performance of concretes made of recycled aggregates from mechanical properties point of view – which is valuable for researchers and practitioners.

I would suggest the following minor suggestions and I hope the authors take these comments into account for the manuscript to improve and be applicable for publication.

1)     What is UEPG?

2)     In Fig. 4, please add the diameter of spilled concrete to each sub-figure.

3)     Please add a discussion on some of the possible techniques that can be used to improve performance of concrete utilizing recycled aggregates.

Author Response

Title:  The effect of fine and coarse recycled aggregates on fresh and mechanical properties of self-compacting concrete

Authors: Mahmoud Nili 1,2*, Hossein Sasanipour1 and Farhad Aslani3,4  

Ref.No.: Article Materials 479119

Dear chief editor,

I appreciate the time and efforts by the editor and referees in reviewing this manuscript. We have addressed all issues indicated in the review report, and believed that the revised version can meet the journal publication requirements.

Response to Comments from Reviewer 2:

Comment 1:

UEPG was explained.

Comment 2:

Diameters of the spread flow of all mixtures in Fig. 4 were added.

Comment 3:

Introduction was edited based on the comment.

Thank you very much for your consideration.

Best Regards,

Mahmoud Nili
